# Exploring the Suitability of Rule-Based Classification to Provide Interpretability in Outcome-Based Process Predictive Monitoring

Suhwan Lee [1] , Marco Comuzzi [2],* and Nahyun Kwon [2]

1 Department of Information and Computing Sciences, Utrecht University, 3584 CC Utrecht, The Netherlands; s.lee@uu.nl
2 Department of Industrial Engineering, Ulsan National Institute of Science and Technology, Ulsan 44919, Korea; eekfskgus@unist.ac.kr
* Correspondence: mcomuzzi@unist.ac.kr

**Abstract:** The development of models for process outcome prediction using event logs has evolved in the literature with a clear focus on performance improvement. In this paper, we take a different perspective, focusing on obtaining interpretable predictive models for outcome prediction. We propose to use association rule-based classification, which results in inherently interpretable classification models. Although association rule mining has been used with event logs for process model approximation and anomaly detection in the past, its application to an outcome-based predictive model is novel. Moreover, we propose two ways of visualising the rules obtained to increase the interpretability of the model. First, the rules composing a model can be visualised globally. Second, given a running case on which a prediction is made, the rules influencing the prediction for that particular case can be visualised locally. The experimental results on real world event logs show that in most cases the performance of the rule-based classifier (RIPPER) is close to the one of traditional machine learning approaches. We also show the application of the global and local visualisation methods to real world event logs.

**Keywords:** business process; event log; predictive monitoring; explainability; rule-based classification

## 1. Introduction

Predictive monitoring of business processes [1,2] has emerged in the last 10 years as a discipline of process mining [3] that focuses on creating predictive models of aspects of interest in business processes. Predictive models exploit information available in so-called event logs, i.e., logs of systems supporting the execution of business processes. Event logs contain events, each belonging to a different execution instance (or *case*) of a process. Events record information, such as the time of recording, a label describing the activity that was executed, an id of the (human) resource in charge of execution and possibly other domain specific attributes.

In addition to the prediction of the next event in a running case [4] or time-based aspects [5], prediction of the outcome of a running process case [6] is one of the main use cases of business process predictive monitoring. In outcome prediction, an event log contains a categorical label, usually binary, specifying the outcome of cases. This label could be given or reconstructed from other information in an event log. For instance, several publicly available event logs contain labels capturing whether process cases satisfy a given declarative constraint predicated on the order and occurrence of certain activities in a case. The objective is then to use event logs to train a predictive model that accurately predicts the outcome of incomplete (running) process cases. As such, outcome prediction is solved using classification techniques.

One of the current main challenges of machine learning predictive models is related to their *interpretability*, that is ([7], p. 1) "the lack of transparency behind their behaviours, which leaves users with little understanding of how particular decisions are made by these models". This concern is obviously relevant also in process predictive monitoring, where the output of predictive models may be used by process owners to take important decisions about specific cases, or even to take decisions about process execution automatically.

There are fundamentally two ways to approach interpretability of machine learning models [7], i.e., interpretable model extraction and post-hoc explanation. The former implies to generate models that are inherently interpretable, such as rule-based models and, to a lesser extent, decision trees. The latter aims at explaining what type of knowledge has been acquired by models during the training, e.g., explaining the values obtained for weights and biases in a neural network in a human-interpretable way [8].

In this paper, we explore the suitability of association rule mining [9] for supporting outcome-based predictive process monitoring. In particular, first we propose a way to pre-process data to make them suitable for the application of rule-based classifiers. Then, to support the model interpretability, we propose different ways to visualise the rules that constitute a rule-based outcome predictive monitoring model. Specifically, we propose two types of visualisations: (i) a way to globally visualise the features and their values generally determining the value of the outcome label, and (ii) a way to visualise the rules that are activated to determine the outcome predicted for a particular case in an event log.

More specifically, from an evaluation standpoint we address the following two problems:

- To compare the performance of a rule-based classifier against traditional classifiers, such as random forests and gradient boosting machines, commonly used in outcome-based predictive process monitoring. The aim in this case is to assess whether the level of performance achievable by the former can be comparable to the one of the latter.
- Exploiting the intrinsic interpretability of rule-based classifiers, to propose visualisations that foster the interpretability of outcome-based process predictive models at the global level, i.e., generally regarding all the traces in an event log, and at the local level, i.e., specifically for a particular trace in an event log.

The proposed framework is evaluated on publicly available event logs. The experimental results show that the performance of a rule-based classifier in outcome-based predictive monitoring is aligned with or only slightly lower than the one of other traditional classification techniques commonly adopted by the literature in most cases. Most importantly, in our evaluation we show how the proposed visualisations can be applied to support the interpretation of the behaviour of an outcome-based predictive model on real-world event logs.

The paper is organised as follows. Section 2 reviews the related work. Section 3 provides some required preliminary notation, while Section 4 formalises the problem of outcome prediction and discusses the proposed approach in detail. Experimental results on real world event logs and examples of visualisations are presented in Section 5 and conclusions are finally drawn in Section 6.

## 2. Related Work

The related work can be classified in the following four areas: (i) outcome-oriented predictive monitoring in process mining, (ii) interpretable (explainable) predictive models in process mining, (iii) association rule mining in process mining, and (iv) interpretable (or explainable) classification with association rule mining.

Regarding outcome-oriented predictive monitoring, Teinemaa et al. [10] provide a general framework for outcome-based process monitoring and an insightful benchmark comparing the performance of different approaches from the perspective of sequence encoding, trace bucketing, and classification algorithms.

Notable approaches in this field are the ones presented by Di Francescomarino et al. [11], which adopts trace clustering before predictive classification on running cases, and Senderovich et al. [12], which adopts inter-case feature encoding to capture case inter-

action scenarios while predicting the remaining time of patient treatments. More recently, Wang et al. [13] have suggested an approach based on deep learning, i.e., bidirectional LSTM with attention mechanism, that outscores other classification algorithm traditionally adopted in this field, such as random forests and gradient boosting machines. Finally, Ferilli et al. [14] have addressed the problem of process outcome prediction in different contexts with a declarative approach that exploits the availability of a formal process model. In summary, previous works in this field have mainly experimented with different classification algorithms and feature encoding techniques to develop more accurate predictive models, rather than focusing on model interpretability.

Regarding interpretable (explainable) predictive models in process mining, Brunk et al. [15] propose a method to predict the next event in a trace based on dynamic bayesian networks, which are an example of inherently interpretable classification techniques. Harl et al. [16] propose to use gated graph neural networks to predict binary process outcomes by transforming an event log into a graph adjacency matrix and feature maps. The trained models in this case is interpretable because each event is scored for its similarity to others in the training set. Therefore, events with higher score are intended to bear a stronger impact on the outcome. Finally, Galanti et al. [17] adopt Shapley value theory to explain factor influence on the outcome of traces predicted using an LSTM model.

Regarding the application of association rules in event log analysis, we highlight the work by Djenouri et al. [18], which applies frequent item-set mining to extract frequent item-sets in an event log that could approximate the relationship among activities normally captured by a process model. Previously, Maggi et al. [19] have applied association rule mining to extract rules that are then used to build process models. Böhmer and Rinderle-Ma [20] propose a trace anomaly detection technique that exploits association rules mined from an event log. The support of association rules, in particular, is used to discriminate between normal and anomalous traces. Ferilli [21] has proposed a first-order logic-based formalism that is more powerful than traditional process modelling techniques and can be used to specify complex conditions (i.e., rules) regarding the process execution.

Association rule mining has already been adapted to create inherently interpretable predictive model in other research field. Mencía et al. [22] propose general methods for multi-label classification using association rule mining. They stress, in particular, that association rules highlight common causal relations in the data, which cater for better explainable predictions. In the health care field, Ji et al. [23] adopted fuzzy rule mining combined with causality rank of infrequent adverse drug reaction to distinguish relationship between target drug and adverse event response probability. In [24], next-day stock price movement as a categorical outcome (e.g., up, down, or hold) has been predicted using association rules extracted from historical stock movements.

In summary, most approaches in the literature for outcome-based predictive process mining are targeting performance improvement. Several recent papers address the interpretability of process predictive models, but only from a post hoc perspective, mainly developing an interpretative stage to explain a deep learning predictive model. In the specific field of business process analysis using event logs, association rule mining has been applied in process model approximation and anomaly detection, but not for obtaining inherently interpretable predictive models, which is the focus of the method proposed in this paper.

## 3. Preliminaries

An event log $EL$ contains events. An event $e$ is a tuple $e = \langle c, \{(d_i, v_i)\}_{i=1,...,D} \rangle$, where $c$ is the case id and $(d_i, v_i)$ are a set of $D$ attributes and their values. We refer to $\mathcal{E}$ as the universe of events, and to $V_i$ as the domain of attribute $d_i$, which can be discrete, $V_i = \{v_{i,j}\}_{j=1,...,J}$, or continuous, $V_i = [v_{i,min}, v_{1,max}] \subseteq \mathbb{R}$. The case id, activity, and timestamp attributes are normally present in all event log. We refer to them using the letters $c$, $a$, and $t$, respectively, and use a dotted notation to reference them when needed, e.g., $e.c$ to refer to the case id of event $e$. For instance, the event

$e = (45, assess, 2020.1.2, Alice, amount = 1000, type = deep)$ captures the fact that, in a process case associated with loan request number 45 ($e.c$), the resource Alice has executed a deep *assessment* ($e.a$) of a loan request of USD 1000 on January 2nd, 2020 ($e.t$).

Any continuous domain $V_i$ can be mapped into a discrete domain $\hat{V}_i = \{\hat{v}_{i,j}\}$ if necessary. For instance, the loan amount in the example above can be mapped into three discrete levels (high, medium, low) according to the amount requested. We, therefore, introduce an attribute-specific discretisation function $disc_i : V_i \longrightarrow \hat{V}_i$, with $disc(v_i) = \hat{v}_{i,j}$.

The sequence of events generated in a given case form a trace $\sigma = [e_1, \ldots, e_n]$, where $\forall i \in [1, n]$, $e_i \in \mathcal{E}$, $\forall i, j \in [1, n]$, $e_i.c = e_j.c$, i.e., all events belong to the same case, and $\forall i \in [1, n-1]$, $e_i.t < e_{i+1}.t$, i.e., events in a trace are ordered in ascending order using the timestamp attribute. The universe of all traces is denoted by $\mathcal{S}$. Note that attributes of events $e_i$ belonging to a trace $\sigma$ may be the same $\forall e_i \in \sigma$. We refer to these attributes as case-level attributes. For instance, the amount requested in a loan request process is a case-level attribute. In the remainder, we refer to $D_E$ and $D_C$ as the set of event- and case-level attributes in an event log $EL$, respectively, and we assume that (i) all events in $EL$ may have a different value for the same event-level attributes $D_E$ and (ii) all events in $EL$ belonging to the same trace have the same values for the case-level attributes $D_C$. Given a trace $\sigma$ and an integer $l < n$, the prefix function $pr$ returns the first $l$ events of $\sigma$, that is, $pr(\sigma, l) = [e_1, \ldots, e_l]$.

## 4. Framework

For applying rule-based classification, we assume that all attributes of events in $EL$ are discrete, that is, continuous attributes in an event log have been discretised if needed. For each attribute $d_i$, we define the attribute itemizer function $it_i^a : \mathcal{D} \longrightarrow \{0, 1\}^J$, which maps an attribute $d_i$ into a sequence of $J$ binary items, obtained by one-hot encoding the value of an attribute within its domain, that is $it_i^a(d_i) = \{dummy_j\}_{j=1,\ldots,J}$, with:

$$dummy_j = \begin{cases} 1 & \text{if } d_i = v_{i,j} \\ 0 & \text{if } d_i \neq v_{i,j} \end{cases}$$

The event itemizer $it^e : \mathcal{E} \longrightarrow \{0, 1\}^n$ maps an event into the set of items derived from its event-level attributes: $it^e(e) = \bigcup_{d_i \in e} it^a(d_i)$, with $d_i \in D_E$.

The trace itemizer $it^t : \mathcal{S} \longrightarrow \{0, 1\}^n$ maps a trace, possibly incomplete, into the set of items derived from its events and from case-level attributes:

$$it^t(\sigma) = it^a(d_1), \ldots, it^a(d_C), it^e(e_1), \ldots, it^e(e_n),$$

where $C$ is the number of case-level attributes in $EL$.

Given a prefix length $L$, we define an item matrix $T^L$ containing, in each row, the items generated by a prefix in an event log:

$$T^L = \begin{bmatrix} it^t(pr(\sigma_1, L)) \\ \vdots \\ it^t(pr(\sigma_N, L)) \end{bmatrix}$$

where $N$ in the number of traces in $EL$.

Let us refer to the columns of $T^L$ as $\mathcal{X}_1, \ldots, \mathcal{X}_P$. Let us also define a labeling function $y : \mathcal{S} \longrightarrow \mathcal{Y}$ mapping a trace $\sigma \in \mathcal{S}$ to its class label $y(\sigma) \in \mathcal{Y}$, with $\mathcal{Y}$ being the domain of the class labels. For outcome predictions, $\mathcal{Y}$ is a finite set of categorical outcomes. Consistently with the literature [10], we consider binary outcomes, i.e., $\mathcal{Y} = \{0, 1\}$. Note that all prefixes generated from a trace $\sigma$ have the same class label.

An outcome classifier $oc : \mathcal{X}_1 \times \ldots \times \mathcal{X}_P \longrightarrow \mathcal{Y}$ is a function mapping the items of a (possibly incomplete) trace into its class label, that is, $oc(it^t(\sigma)) = y$.

A machine learning-based (ML-based) classifier (*mlbc*) is an *oc* developed using machine learning classification techniques, such as a decision tree or an artificial neural

network. In other words, an ML-based classifier uses the items $\mathcal{X}_p$ as features to train a model that predicts the outcome label of traces. The literature typically has focused on this type of techniques to solve the outcome prediction task in predictive monitoring [25].

An association rule mining-based (ARM-based) classifier (*armb*) is an *oc* developed using association rule mining techniques, such as RIPPER [26].

In this paper, we tackle the following two problems:

- To compare the performance of ML-based classifiers and an ARM-based classifier, in order to assess whether the level of performance achievable in outcome-based process predictive monitoring by rule-based classification is at least comparable with the one achieved by other techniques in the literature;
- Exploiting the intrinsic interpretability of rule-based classifiers, to propose visualisations that foster the interpretability of outcome-based process predictive models at the global level, i.e., generally regarding all the traces in an event log, and at the local level, i.e., specifically for a particular trace in an event log.

To tackle the first objective, it is necessary to pre-process an event log into a discrete set of items, as discussed above.

A first data preparation step, therefore, concerns the discretisation of the continuous attributes in an event log. The values of attributes with continuous domains are clustered into three groups, usually labelled 'small', 'medium', and 'large', using the 33% and 66% percentiles as thresholds, e.g., a value is mapped to the categorical value 'medium' if it falls in a percentile comprised between the 33% and 66%.

Next, to train and test ML-based predictive models, an event log is pre-processed using prefix-length bucketing and index-based encoding [27]. That is, we create a different model for each prefix length, trained/tested using prefixes pre-processed to obtain a fixed-length features vector. All attributes of an event log are pre-processed using one-hot encoding.

Figure 1 shows an example of pre-processing an event log where the continuous attributes are first discretised and then attributes are one-hot encoded to obtain an item matrix. The figure shows that the categorical attributes, such as activity labels, are one-hot encoded index-based, i.e., for each position in a trace, an item is generated for all possible activities in the event log, and only the item corresponding to the activity that was executed is set to 1. The encoding of the numerical attributes (attributes C2 and timestamps in the example) occurs through the discretisation described above. Finally, all the items obtained are integrated into the item matrix that is used for the classification task.

For ARM-based models, the objective of pre-processing is to obtain an extended item matrix from which rules can be extracted. Similarly to the case of ML-based models, we consider prefix-length bucketing and index-based encoding, that is, we create different item matrices for each prefix length.

| Case ID | Activity | Timestamp | C1 | C2 | E1 | E2 | Label |
|---|---|---|---|---|---|---|---|
| 1 | A | 2010-10-12 07:00:00 | G | 84 | Empty | 10 | 1 |
| 1 | B | 2010-10-12 18:57:34 | G | 84 | False | | 1 |
| 2 | A | 2010-10-12 18:57:58 | O | 1698 | Empty | 15 | 0 |
| 2 | C | 2010-10-12 19:00:23 | O | 1698 | Empty | | 0 |
| 3 | E | 2010-10-28 18:29:56 | G | 126 | False | | 1 |
| 3 | B | 2010-10-28 18:30:03 | G | 126 | Empty | | 1 |

<Original event log>

| Case ID | Activity1 _A | Activity1 _E | Activity2 _B | Activity2 _C | E1_1 _Empty | E1_1 _False | E1_2 _Empty | E1_2 _False | E2_1 _small | E2_1 _medium | E2_1 _large | Label |
|---|---|---|---|---|---|---|---|---|---|---|---|---|
| 1 | 1 | 0 | 1 | 0 | 1 | 0 | 0 | 1 | 1 | 0 | 0 | 1 |
| 2 | 1 | 0 | 0 | 1 | 1 | 0 | 1 | 0 | 0 | 1 | 0 | 0 |
| 3 | 0 | 1 | 1 | 0 | 0 | 1 | 1 | 0 | 0 | 0 | 0 | 1 |

<Event attributes categorization>

| Case ID | T1_small | T1_medium | T1_long | T2_small | T2_medium | T2_long | Label |
|---|---|---|---|---|---|---|---|
| 1 | 0 | 1 | 0 | 1 | 0 | 0 | 1 |
| 2 | 1 | 0 | 0 | 1 | 0 | 0 | 0 |
| 3 | 0 | 0 | 1 | 0 | 1 | 0 | 1 |

<Timestamp categorization>

| Case ID | C1_G | C1_0 | C2_small | C2_medium | C2_large | Label |
|---|---|---|---|---|---|---|
| 1 | 1 | 0 | 1 | 0 | 0 | 1 |
| 2 | 0 | 1 | 0 | 0 | 1 | 0 |
| 3 | 1 | 0 | 0 | 1 | 0 | 1 |

<Case attributes categorization>

| Case ID | C1_G | C1_O | C2_ small | C2_ medium | C2_ large | Activity1 _A | Activity1 _E | Activity2 _B | Activity2 _C | T1_ small | T1_ medium | T1_ long | T2_ small | T2_ medium | T2_ long | E1_1_ Empty | E1_1_ False | E1_2_ Empty | E1_2_ False | E2_1_ small | E2_1_ medium | E2_1_ large | Label_1 | Label_0 |
|---|---|---|---|---|---|---|---|---|---|---|---|---|---|---|---|---|---|---|---|---|---|---|---|---|
| 1 | 1 | 0 | 1 | 0 | 0 | 1 | 0 | 1 | 0 | 0 | 1 | 0 | 1 | 0 | 0 | 1 | 0 | 0 | 1 | 1 | 0 | 0 | 1 | 0 |
| 2 | 0 | 1 | 0 | 0 | 1 | 1 | 0 | 0 | 1 | 1 | 0 | 0 | 1 | 0 | 0 | 1 | 0 | 1 | 0 | 0 | 1 | 0 | 0 | 1 |
| 3 | 1 | 0 | 0 | 1 | 0 | 0 | 1 | 1 | 0 | 0 | 0 | 1 | 0 | 1 | 0 | 0 | 1 | 1 | 0 | 0 | 0 | 0 | 1 | 0 |

<Merged pre-processed event log>

**Figure 1.** Example of pre-processing an event log.

### 4.1. Outcome Prediction

As ML-based models, we consider gradient boosting machines (XGB) and random forests (RF). Both classifiers have shown to perform well in event log predictive monitoring use cases [12].

As ARM-based model, we consider the Repeated Incremental Pruning to Produce Error Reduction (RIPPER), which has been originally proposed to improve the performance of the *IREP* model [28]. RIPPER generates a classification rule by (i) splitting the samples randomly into two disjoint subsets, i.e., a growing set and a pruning set, and (ii) generating classification rules using an information gain-based algorithm. Once a rule is generated, it is immediately pruned by repealing any final sequence of conditions. RIPPER is based on the principle that classification rules can be grown until a positive information gain is achieved. As an ARM-based classifier, RIPPER is generally considered better performing than other rule-based classifiers deriving the classification rules from a trained decision tree. The details about the hyper-parametrisation of the ML- and ARM-based models are given later before introducing the experimental results.

*4.2. Visualisations to Support Model Interpretability*

The last part of the proposed framework is an approach for visualising the rules used for outcome prediction in a more interpretable way. Although association rules are inherently interpretable and a basic interpretation of rules can be operated domain experts by skimming through a printed list of rules, having a means to visualise effectively the rules can dramatically increase their interpretability by human decision makers [29].

The rules defining an outcome-based process predictive monitoring model can be visualised at two levels:

- *Global level.* At the global level, the objective is to give to decision makers an idea of which features of the data define the behaviour of a predictive model. Therefore, at the global level we aim at providing an interpretable way to quickly understand the components of the rules of the predictive model, i.e., the features involved in the antecedents and consequents of the rules.
- *Local level.* In explainable outcome-based predictive monitoring, the local level refers to interpreting the behaviour of a predictive model for an individual process trace [17]. Hence, we propose a way to visualize, for a single trace at a given prefix, the rules that are activated to obtain a prediction and the features of the trace that trigger the activation.

*Global level visualisation.*

At the global level, the visualisation of rules that we propose is based on the following principles:

- A network graph is built for one event log at a given prefix length for each label, i.e., considering the rules generated by the model trained with the prefixes at that prefix length and for which the prediction evaluated to that particular label (0 or 1);
- Owing to the fact that each graph represents all the rules for a given label, each graph is built only with information related to the antecedents of the rules; in particular, each node in the graph is an item (that is, a feature associated with a possibly discretised value) that appears in at least one rule and edges connect two items (nodes) that appear together in the antecedent of a rule;
- The thickness of nodes represents the frequency of appearance of an item in the rules relative to the total number of items in the antecedents of the rules;
- To facilitate the interpretation and the identification of items, four types of items are identified and different colours are assigned to the corresponding nodes in a graph: items derived from an *activity* label of an event, from *timestamps*, from other *case-level attributes*, or from other *event-level attributes*;
- The thickness of the edges in the graph represents the relative frequency at which the two nodes linked by an edge appear together in one rule, relative to the total number of rules.

*Local level visualisation.*

At the local level, the objective is to understand the rules that have been activated to obtain the value of the predicted label, emphasising in particular the attributes that triggered the activation of the rules.

The local level visualisation consists of 4 visual elements:

1. Case information;
2. Case-level attribute information;
3. Trace-level attribute information;
4. Association rules.

The case information elements shows the case id, the predicted outcome, and the actual outcome of a case to which the trace (prefix) for which a prediction has been obtained belongs. The case- and trace-level attribute elements show the values assumed in the trace by the features of the predictive model derived from the event log attributes. Finally,

the last element shows the association rules of the model, highlighting (in blue) the ones that have been activated to obtain the prediction, i.e., the ones for which the antecedents evaluate to 1. Note that the attributes responsible for the activation of the rules are also highlighted using the same colour.

In summary, the local-level visualisation gives an overview of the rules that have been activated in the context of the prefix under scrutiny, highlighting also which event attributes have contributed to the activation of the rules. This is intended to help decision makers making sense of the behaviour of the predictive model, i.e., understanding which feature values are responsible for a given prediction.

Concrete examples of both types of visualisations for predictions obtained on prefixes of real world event logs are presented in the next section.

## 5. Evaluation

Section 5.1 presents the datasets used in our evaluation, along with the hyper-parameters of ML-based classifiers and the parameter values used for the RIPPER (ARM-based) classifier. The performance comparison between rule-based and traditional outcome predictive models in presented in Section 5.2, while Section 5.3 presents and discusses examples of global- and local-level visualisations applied to the real world event logs considered in the experimental evaluation.

### 5.1. Datasets, Model Parameters, and Hyper-Parameters

We consider six event logs publicly available at https://data.4tu.nl/ (accessed on 27 April 2022) published by the Business Process Intelligence Challenge in 2011, 2015 (Note that there are five different instances of this event log, with data from different municipalities. We consider the instance numbered as 1, 2, and 3), 2012 and 2017. The BPIC 2011 event log is about a diagnosis and treatment process in a Dutch academic hospital. The BPIC 2015 event logs are from a process of managing building permit requests at different Dutch municipalities. The BPIC 2012 and BPIC 2017 event logs are from a process of managing loan requests at a Dutch financial institution. These logs have been chosen because they contain an outcome label and have been used by previous research on outcome-based process predictive monitoring [10]. Table 1 shows the descriptive statistics of these event logs. Note that event logs differ widely for number of activities, number of attributes, and label class proportion.

**Table 1.** Descriptive statistics of event logs used for evaluation.

| | # Activities | # Cases | # Variants | # Categorical Case Attributes | # Continuous Case Attributes | # Categorical Event Attributes | # Continuous Event Attributes | # Outcome 0 Cases | # Outcome 1 Cases |
|---|---|---|---|---|---|---|---|---|---|
| BPIC 2011 | 677 | 1143 | 981 | 4 | 6 | 5 | 0 | 683 | 460 |
| BPIC 2015_1 | 289 | 1199 | 1100 | 8 | 1 | 3 | 0 | 910 | 289 |
| BPIC 2015_2 | 304 | 832 | 828 | 8 | 1 | 3 | 0 | 670 | 162 |
| BPIC 2015_3 | 277 | 1409 | 1349 | 8 | 1 | 3 | 0 | 1122 | 287 |
| BPIC 2012 | 36 | 13,087 | 4366 | 1 | 1 | 2 | 0 | 10,844 | 2243 |
| BPIC 2017 | 26 | 31,509 | 4047 | 1 | 5 | 2 | 0 | 14,281 | 17,228 |

In the BPIC 2011 and BPIC 2015 event logs, given a linear temporal logic constraint predicated on the order and occurrence of activities, the outcome label is set to 1 for cases satisfying the constraint and to 0 for other cases. In the BPIC 2015 event logs, the constraint is that activity 'send confirmation receipt' is eventually followed by the activity 'retrieve missing data'. In the BPIC11 event log, the constraint is that either the activity 'ca-19.9 tumormarker' or 'ca-125 mbv meia' occur in a trace. In the BPIC 2012 and BPIC 2017 event logs, the outcome label captures whether a loan request is eventually accepted or not.

The value of hyper-parameters has a great influence on the performance of a model. To improve the performance of the model, we tune the hyper-parameters with their range using Bayesian Optimisaion. These hyper-parameters and search spaces are listed in Table 2.

**Table 2.** Search space of hyper-parameters for the models used in the evaluation.

| # Model | # Parameter | # Range |
|---|---|---|
| RF | n estimators | [100, 200, 300] |
| | max features | ['auto', 'sqrt'] |
| | max depth | (10, 1000) |
| | min samples split | [2, 6, 10] |
| | min samples leaf | [1, 4, 7] |
| | bootstrap | [True, False] |
| XGB | n estimators | [100, 200, 300] |
| | learning rate | [0.1, 0.3, 0.5] |
| | subsample | [0.3, 0.5, 0.7] |
| | max depth | (10, 1000) |
| | colsample bytree | [0.5, 0.6, 0.7] |
| | min child weight | [1, 2, 3, 4] |
| RIPPER | prune size | [0.33, 0.5] |
| | k | [1, 2] |

We consider prefixes between 5 and 40 with a gap of 5 for all event logs. For example, if prefix length is equal to 5, then RF, XGB, and RIPPER are trained and tested on a dataset containing all the prefixes of length 5 in the event log, that is, partial traces considering the first 5 events of traces with at least 5 events. To cope with sampling bias, we consider five-fold cross-validation. The proportion of train and test set sampling in every replication of the experiment is 70% to 30%. The code to reproduce the experiments is publicly available at https://github.com/eekfskgus/Rule_based_classification_for_interpretability (accessed on 27 April 2022).

*5.2. Results and Discussion*

The objective of the evaluation of the proposed approach is two-fold. First, our aim is to appraise whether the performance of the ARM-based classifier is at least comparable with the one of other ML-based approaches in outcome-based predictive process monitoring. If that were not to be the case, in fact, it would not make sense to employ ARM-based classifiers in this prediction task, even if the model obtained were intrinsically explainable. Second, our aim is to demonstrate the applicability of the proposed local and global level ARM-based model visualisation to real world event logs.

Regarding the first objective, the performance of the proposed ARM-based classifiers compared against one of ML-based classifiers is shown in Figure 2. Note that the figure shows the value of different performance measures commonly used for classification tasks by prefix length.

First, we note that, when the AUC measure is considered, the performance of the ARM- and ML-based classifier is comparable at all prefix lengths and for all datasets, with the only exception of the BPIC 2011 event log at a higher prefix length. AUC is the principal measure of performance normally considered by previous work in outcome-based predictive monitoring [10]. As far as precision is concerned, RIPPER achieves, in general, high precision on all event logs. In some cases (BPIC 2011 and BPIC 2012 event logs at low prefix lengths), the precision achieved by RIPPER is higher than ML-based models. This is also true for the BPIC 2017 event log. However, RIPPER achieves, at the same time, consistently lower accuracy with this event log.

For some event logs (BPIC 2012, but most evidently BPIC 2017), the recall achieved by RIPPER is low, which affects, also, the F-score performance. For the BPIC 2011 event log, the performance of RIPPER is comparable with ML-based classifiers only at low prefix lengths (up to 20). These peculiarities of the results may be due to intrinsic characteristic of the datasets and/or the business process generating them. For instance, the BPIC 2012 and 2017 event logs are extracted from the execution of the same business process (loan requests at a financial institution), but using two different information systems to support the execution (this system has been upgraded before the BPIC 2017 event log was captured).

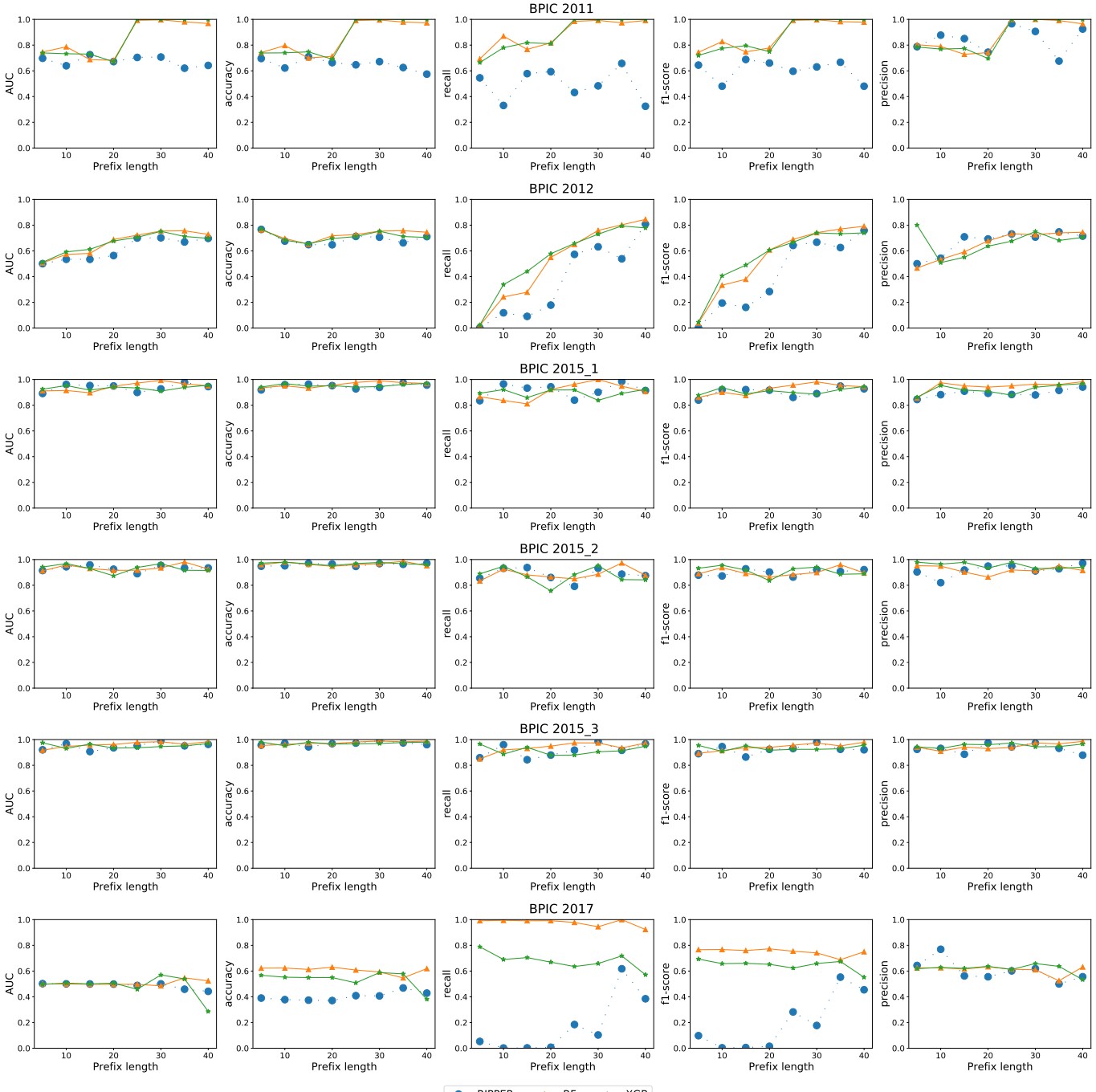

**Figure 2.** Predictive performance of RIPPER against ML-based models.

Based on the results shown in Figure 2, in particular the analysis of the AUC results, we can claim that the ARM-based classifier RIPPER is not particularly worse than RF or XGB for this classification task. Therefore, decision-makers may decide to choose using an ARM-based classifier, since the model obtained is intrinsically explainable, as demonstrated by the proposed visualisations discussed next.

### 5.3. Examples of Model Visualisations

Figure 3 shows an example of the global-level visualisations obtained for the BPIC2015 event log at prefix length 10. Several insights regarding the behaviour of the model trained and tested using this event log can be drawn from this visualisation. A first quick comparison of the two graphs clearly reveals that the features determining the positive label (label 1)

differ substantially from the ones determining the negative label (label 0). It appears that, for the positive label, there are basically three groups of features determining the positive outcome: (i) a first cluster involving the co-occurrence of the length of the eighth event to be high (`Time8=Long`) and the activity label of the seventh event (`Activity7`) to be `enter send date retrieving missing data`; (ii) a second cluster involving the occurrence of specific values for the activity label of the seventh event and the case-level attribute `parts`; and (iii) a third cluster involving a larger combination of values of several features. Regarding the latter, the occurrence of the activity `enter send date acknowledgement` in the second position in a case appears to be a discriminant on the label: when this happens, the outcome is in fact more likely to be positive. There is also a fourth cluster of rule antecedents involving the features `SUMleges` and `requestComplete`, which are both features derived from case-level attributes. The negative label global-level model visualisation shows only one cluster involving a high number of features. Considering the number of rules in which the features are involved, i.e., the size of the nodes in the graph, the strongest predictors of the negative outcome appear to be the non-occurrence of the activity `enter send date acknowledgement` as the second event in cases for which the case-level attribute `requestComplete` evaluates to `True`.

Figure 4 shows an example of the global-level visualisations obtained for the BPIC 2017 event log. Given the nature of the prediction for this log, it should not surprise that most features appearing in these visualisation derive from event-level attributes (purple nodes). These in fact often capture the characteristics of the loan request, e.g., the loan amount requested (`OfferedAmount`) or the amount of the monthly instalments (`MonthlyCost`). More in detail, for the negative label, it can be noted that there are two clusters of rule antecedents: the first cluster (left hand side) involves requests rejected characterised by a medium amount requested, while the second one (right hand side) applies to loans for which the amount requested is high. For instance, high amount loan requests appear to be rejected when the number of instalments (`NumberofTerms`) is medium and the monthly instalments (`MonthlyCost`) are not large.

Figure 5 and 6 show examples of the local-level visualisations obtained for traces where the predicted outcome matched the actual one. In both examples, it is possible to clearly identify the rules that have been activated to support the prediction of the positive label. Specifically, Figure 6 shows a case in which the positive label is predicted. Based on the global level explanation, this case has a predictive positive label because it has the case-level features defining the fourth cluster discussed above, i.e., derived from case attributes `SUMLeges` and `requestComplete` (which activate Rule 1). However, the local-level explanation also shows many additional rules that are activated and that may help explaining the positive label. An analysis of these rules can be used by a domain expert decision-maker to better characterise the particular case at hand, deciding for instance whether the positive level predicted should be trusted or not in a specific decision-making context. For instance, for the trace of the event log BPIC 2015_1 depicted in Figure 6, we can see that the short duration of the sixth event (`Time6` in Rule 5), the identity of the resource executing the ninth activity (`Resource9` in Rule 7) or the activity executed at the ninth event (`Activity9` in Rule 8) are feature values important to determine a positive and correct predicted outcome for this particular trace.

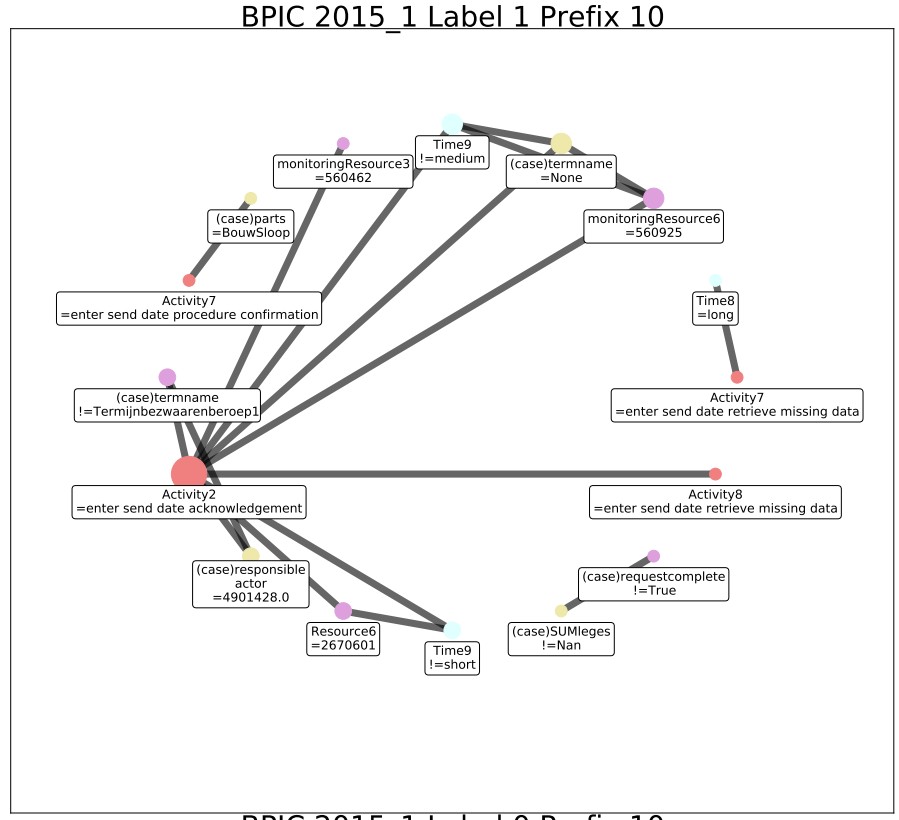

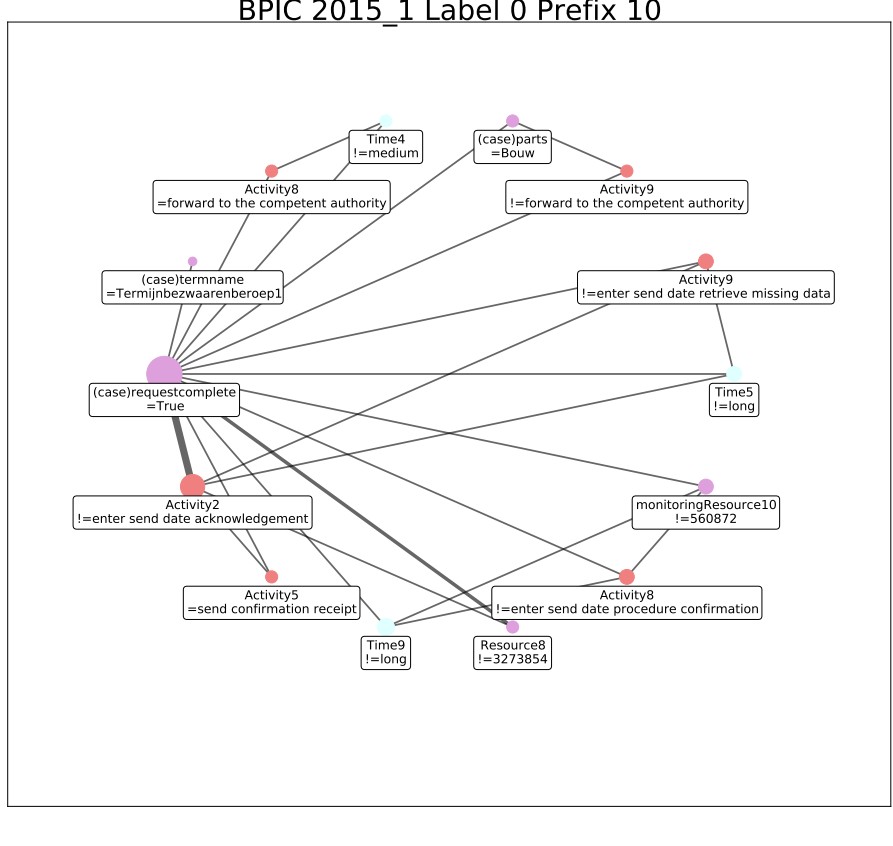

**Figure 3.** Example of global-level model visualisation (BPIC2015_1 event log at prefix 10).

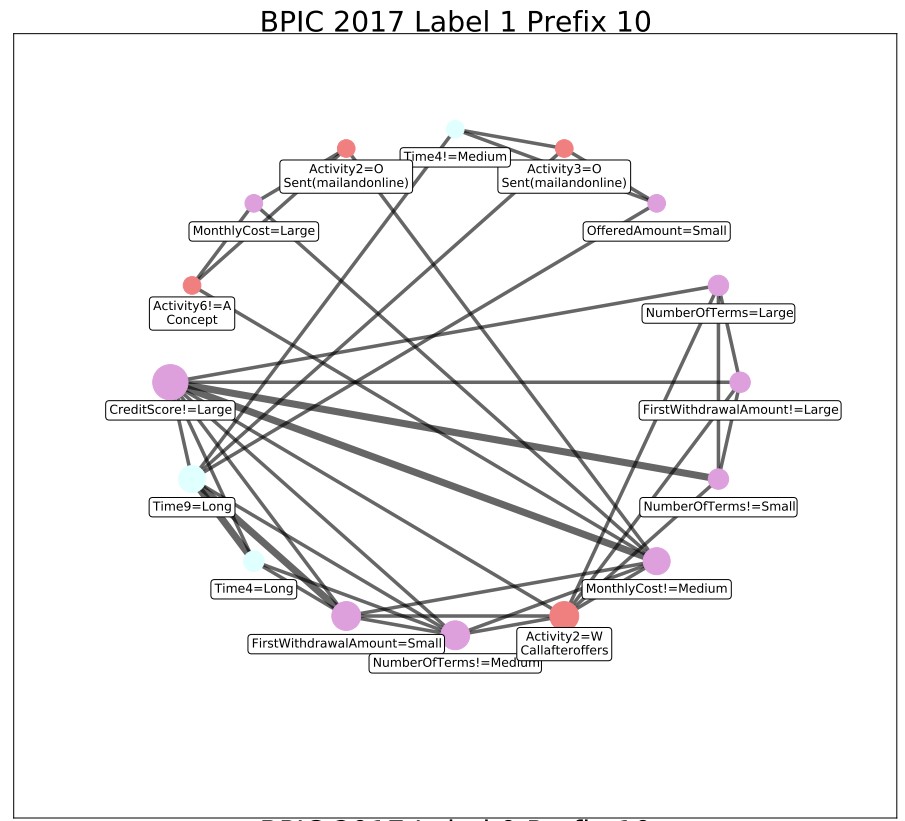

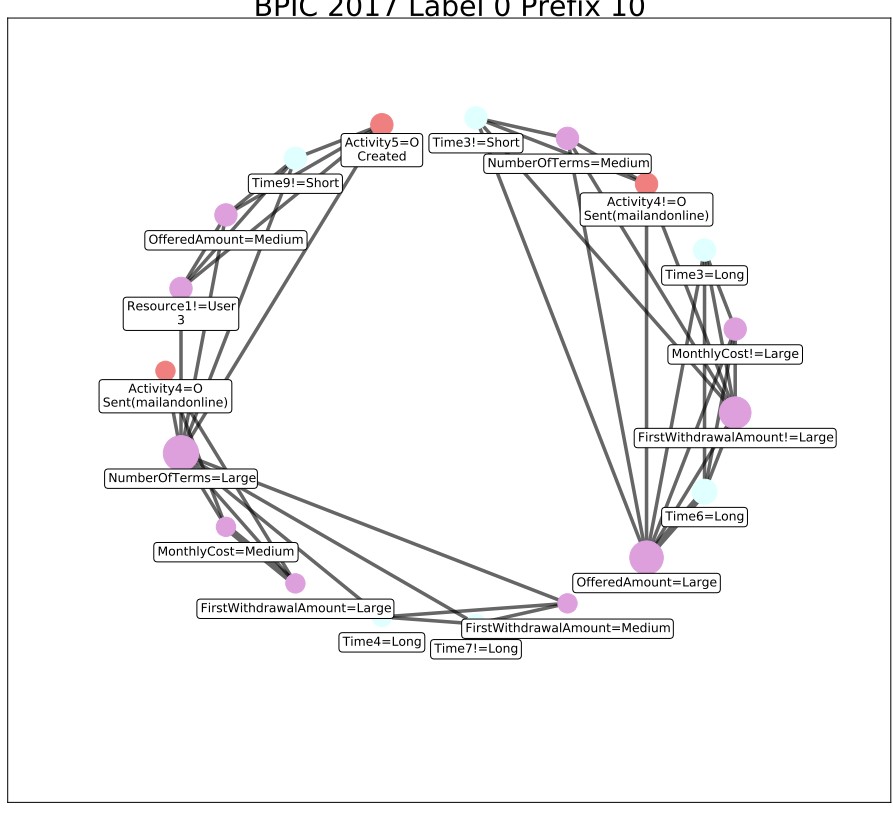

**Figure 4.** Example of global-level model visualisation (BPIC2017 event log at prefix 10).

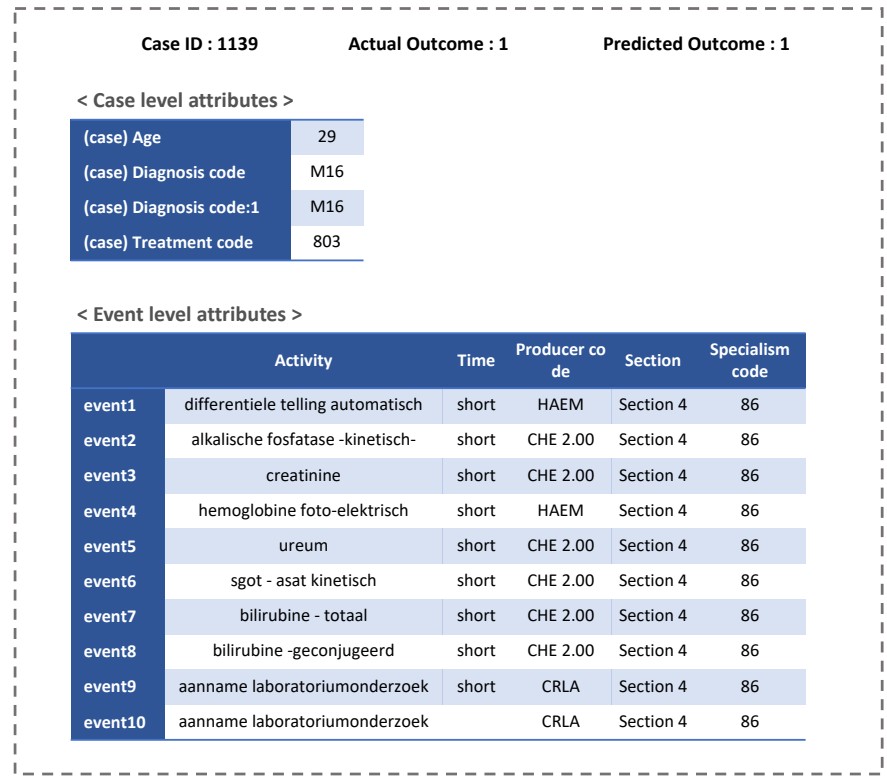

| | Contents | Execution |
|---|---|---|
| **Rule 1** | { Specialismcode3 = '86' } & { (case)diagnosis code = 'M16' } | **Yes** |
| **Rule 2** | { Specialismcode1 = '86'} & { Producercode1 != 'CRLA' } & { (case) Diagnosis code != 'M11' } & { (case) Diagnosis code != 'M13' } & { (case)specialismcode:1 != '7.0' } & { (case) Diagnosis code != '106' } & { (case)treatmentcode:3 != '23.0' } & { (case) Diagnosis code != 'M12' } | **Yes** |
| **Rule 3** | { (case)specialismcode:1 != '7.0' } & { (case) Diagnosis code = 'M14' } & { Specialismcode1 = '86' } | **No** |
| **Rule 4** | { (case) Diagnosis code:1 = 'M14' } & { Producercode1 = 'SGNA' } | **No** |
| **Rule 5** | { Specialismcode1 = '86' } & { (case) Diagnosis code = 'M14' } | **No** |
| **Rule 6** | { (case) Diagnosis code:1 = 'M16' } & { Specialismcode8 = '86 } | **Yes** |
| **Rule 7** | { (case)specialismcode:1 != '7.0' } & { (case) Diagnosis code = 'M14' } & { Producercode2 = 'CHE2' } | **No** |
| **Rule 8** | { (case) Diagnosis code:1 = 'M16' } & { Section7 = 'Section4' } & { Time2 != 'long' } | **Yes** |
| **Rule 9** | { (case) Diagnosis code:1 = 'M16' } & { Producercode8 != 'SGNA' } | **Yes** |

**Figure 5.** Example of local-level model visualisation (a trace of the BPIC 2011 event log at prefix 10).

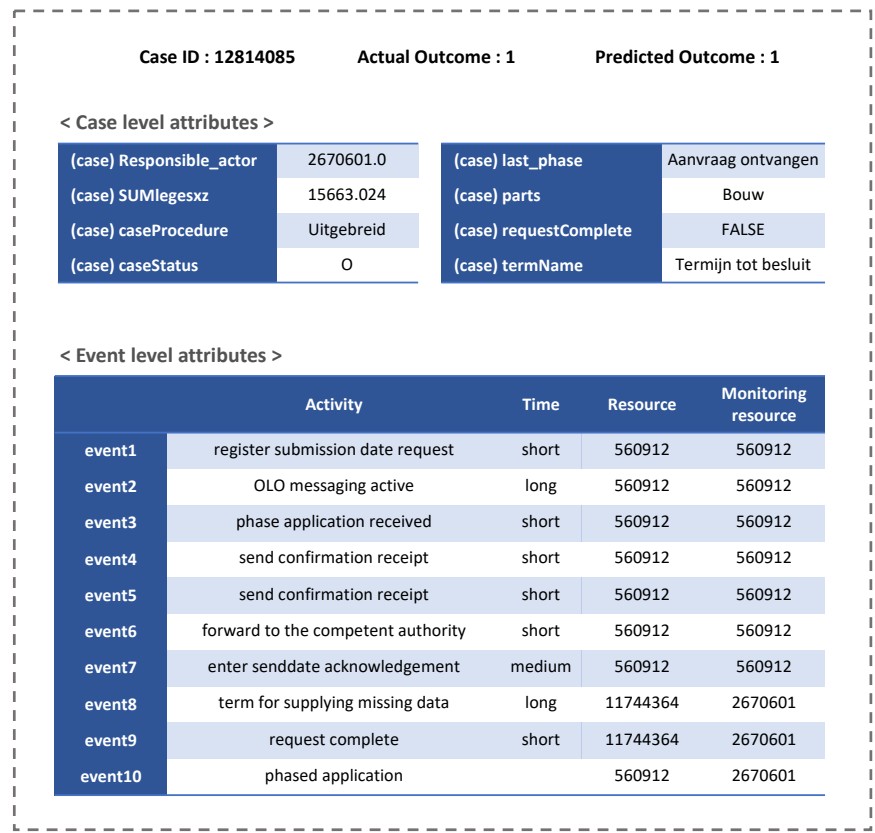

**Label 1 Rules of RIPPER**

| | Contents | Execution |
|---|---|---|
| Rule 1 | { (case)requestComplete = 'False ' } & { (case)SUMleges != 'Nan ' } | Yes |
| Rule 2 | { (case)requestComplete = 'False ' } | Yes |
| Rule 3 | { Time1 = 'short ' } & { Activity7 = 'enter senddate procedure confirmation ' } & { (case) Responsible actor = '4901428.0 ' } | No |
| Rule 4 | { (case) requestComplete = 'False ' } & { (case)responsible_actor = '4901428.0 ' } | No |
| Rule 5 | { (case) requestComplete = 'False ' } & { Time6 != 'long ' } | Yes |
| Rule 6 | { Time1 = short} & { monitoringResource3 = 560462 } | No |
| Rule 7 | { Resource9 != '3273854' } & { (case)termname = 'Termijn tot besluit' } | Yes |
| Rule 8 | { (case) requestComplete = 'False' } & { Activity9 != 'send confirmation receipt' } | Yes |
| Rule 9 | { Time1 = 'short '} & { (case)caseprocedure = 'Uitgebreid' } | Yes |
| Rule 10 | { Activity8 = 'no permit required for application' } | No |

**Figure 6.** Example of local-level model visualisation (a trace of the BPIC 2015 event log at prefix 10).

## 6. Conclusions

This paper has introduced a novel, interpretable approach to outcome-based predictive process monitoring that uses association rule mining. The framework defines how to pre-process an event log to obtain items and has proposed an effective way of visualising the rules obtained using network graphs.

The proposed framework has been evaluated, all other conditions being equal, against predictive models built that use extreme gradient boosting machine and random forest. The performance obtained by the ARM-based classifier is in most cases comparable to the one of the ML-based traditional classifiers. The proposed global- and local-level visualisation help to interpret the features and their respective value that most influence a prediction generally and specifically in a running trace, respectively.

Future work will pursue multiple perspectives, in terms of both application and extension. First, we plan to apply the proposed framework to the more general use case of classification. That is, we intend to use our framework with data different than event logs, studying, in particular, to what extent the type of phenomenon that the data represent impacts the applicability and performance of our framework. Regarding extension, we are planning to develop different visualisation techniques and evaluate them in respect of usefulness and ease of use with pseudo-expert decision-makers, such as university students of BPM subjects. To improve the performance of the framework, in particular to being able to handle information loss during the categorisation process, we are also planning to experiment with fuzzy association rules, which often have been demonstrated to be more reliable than crisp rules for supporting decision making in complex scenarios. Regarding the encoding, we will consider finer grained intervals for discretising continuous attributes, which may affect the performance and the visualisations. The effects of finer grained discretisation on the visualisations, however, can be appreciated only by users with a deep knowledge of the domain where the process takes place. Therefore, this should be evaluated with case studies involving domain experts.

**Author Contributions:** Conceptualization, methodology and writing S.L. and M.C.; software and validation, S.L. and N.K. All authors have read and agreed to the published version of the manuscript.

**Funding:** This work is supported by the 0000 Project Fund (Project n. 1.220047.01) of UNIST (Ulsan National Institute of Science & Technology).

**Institutional Review Board Statement:** Not applicable.

**Informed Consent Statement:** Not applicable.

**Data Availability Statement:** The experiments discussed in this paper can be replicated by using the code and data at https://github.com/eekfskgus/Rule_based_classification_for_interpretability (accessed on 27 April 2022).

**Conflicts of Interest:** The authors declare no conflicts of interest.

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
