# Peer review of "Exploring the Suitability of Rule-Based Classification to Provide Interpretability in Outcome-Based Process Predictive Monitoring"

_algorithms, doi:10.3390/a15060187_

Round 1

Reviewer 1 Report

This work makes a series of contributions to studying how rule-based classifiers perform as interpretable methods for predictive monitoring of processes.

The paper is very well written, easy to read, and the abundant formalization is well presented and seems sound.

The experiments are well justified and designed, pertaining data and metrics.

The results presentation and analysis are appropriate and focused in the claimed contributions.

A few minors spotted:
- In bottom tables of Figures 4 and 5: you could add a missing blank after "(case)" in the respective cells for the Contents column in some of the rows

Author Response

See file attached

Reviewer 2 Report

This submission is about the application of association rules in outcome-based predictive process monitoring. Therein the authors propose different visualisations of rules to obtain interpretable predictive models. Their approach was evaluated on an open accessible set of real world event logs.

Section 4 Framework

  • I think it would help if the two problems listed in this section have been generally introduced in the introduction section as well. I have seen that this has been done in general, only it would be nice to highlight this a bit more compactly in chapter 1.
  • Table 1 is very small nearly unreadable. Same holds true for Figure 3.
  • Typo in line 265 (teh à the)

Section 5 Evaluation

  • Figure 3 is the only figure of the graph-based visualization. The figures are very small and hardly readable. I would have appreciated to show more visualization.
  • Line 347 typo: usng à using

In general, the research question posed is relevant, challenging and well-motivated. The overview of the state of the art is detailed. The structure of the paper is very well structured and the methodology used is sound. The authors have linked their git repository to ensure reproducibility, which I am very pleased about.

Is there a plan to do a user-centred evaluation that tests the comprehensibility of the visualization? That would certainly be very interesting. 

Author Response

See file attached

Reviewer 3 Report

An interesting and overall well-written paper.
There are some technical/formal flaws to be fixed.
The discussion of the experimental outcomes should be expanded.

DETAILED COMMENTS

Concerning related work, the authors might want to take a look at the work by Ferilli et al. on process prediction. It seems to be relevant, at least for a comparison in the related work, since it is based on a declarative approach and it can also use rules. Its application to process outcome prediction in several contexts is described in:
 - S. Ferilli, F. Esposito, D. Redavid & S. Angelastro. Language Identification as Process Prediction using WoMan. In Digital Libraries and Archives – IRCDL 2017, Revised Selected Papers, Communications in Computer and Information Science 733, 159-172, Springer, 2017.
 - S. Angelastro, B.N. De Carolis & S. Ferilli. Predicting User Preference in Pairwise Comparisons Based on Emotions and Gaze. In Advances and Trends in Artificial Intelligence. From Theory to Practice, Lecture Notes in Computer Science 11606, 253-261, Springer, 2019.
 - S. Ferilli, D. Redavid & S. Angelastro. Mining Chess Playing as a Complex Process. In NFMCP 2016: New Frontiers in Mining Complex Patterns, Lecture Notes in Artificial Intelligence 10312, 248-262, Springer, 2017.

while perhaps less relevant to this paper are its use for activity prediction:
 - S. Ferilli & S. Angelastro. Activity Prediction in Process Mining using the WoMan Framework. Journal of Intelligent Information Systems, 53(1):93-112, 2019.
(activity prediction)
and its use of rules:
 - S. Ferilli. Handling Complex Process Models Conditions Using First-Order Horn Clauses. In Rule Technologies – Research, Tools, and Applications, Lecture Notes in Computer Science 9718, 37-52, Springer, 2016.

Notation and formalism is sometimes sloppy:
 - In the event itemizer, by taking the union of the attribute itemizer, how to distinguish the attributes having the same number of possible values or - even worse - the same values for these attributes? In fact, set union cannot distinguish equal values, nor the "role" of each value.
In fact, this is also inconsistent with the use of the items as rows in table T^L, exactly because the ordering is lost (and, for what matter, also the cardinality, i.e., the number of columns might not be the same for all rows).
 - Symbol N in table T^L is not defined.
 - p. 164: shouldn't the range of function oc be Y, not S?
 - p. 165: oc(item^t(\sigma)) it was it^t

Is the discretization into always 3 bins sensible and useful for all domains and attributes? If (as I guess) it is not, what is the loss in expressive power? How (much) does it affect the final outcome?

More comments to Figure 1 would be welcome.

In the BPIC11 and BPIC 2015 datasets, the class is determined by a very specific situation occurring in the traces. As I understand it, also in BPIC 2012 and 2017 the loan request acceptance corresponds to an explicit activity occurring in the cases. Isn't this something that may make the problem easier? What about the (more comple and perhaps more relevant) cases in which the overall process behavior is to be cast in a class without this being associated to specific and explicit clues in the cases?

Discussion and comments on the experimental evaluation in Sec. 5.2 should be expanded. E.g.:
 - p. 8: "For some event logs (BPIC 2012 and BPIC 2017), the recall achieved by RIPPER is low," but recall and F-measure are not shown in Figure 2.
 - Accuracy and Precision for dataset BPIC 2017 in Fig. 2 have a more peculiar behavior that should be commented in the text.
 - Sec. 5.2: In addition to the discussion about the clusters in Fig. 3, wouldn't a discussion of the specific links in the clusters help gaining more understanding about the classes? And what about a comparison of the nodes/edges appearing or not appearing in both labels, along with their associated relevance?
 - Figg. 4-5: a discussion/comparison between the rules found and the "clues" determining the classes, as previously stated, would be interesting and relevant.

PRESENTATION ISSUES

Some typos or grammatical errors must be fixed. E.g.:
 - p. 49-50: "we propose different ways to visualise the rules that constitutes" -> constitute
 - p. 104: "in other research field" -> fields
 - p. 122: "We refer E as" -> to E
 - p. 125: "all event log" -> logs
 - p. 182: "The tackle the first objective" -> to tackle
 - p- 265: "of teh predictive model" -> the
 - p. 266: "the last element show" -> shows
 - p. 295: "given a temporal linear temporal logic" -> linear temporal logic

Figure 2: column headings "AUC", "Accuracy" and "Precision" should be at the top, above "BPIC2011". Also, since these are actually the labels for the y axis, they clash with the labels for the x axis being in the same columns.

Author Response

See file attached

Round 2

Reviewer 3 Report

ok